# *Drimia indica*: A Plant Used in Traditional Medicine and Its Potential for Clinical Uses

**DOI:** 10.3390/medicina55060255

**Published:** 2019-06-07

**Authors:** Sonali Aswal, Ankit Kumar, Ruchi Badoni Semwal, Ashutosh Chauhan, Abhimanyu Kumar, Jörg Lehmann, Deepak Kumar Semwal

**Affiliations:** 1Research and Development Centre, Faculty of Biomedical Sciences, Uttarakhand Ayurved University, Harrawala, Dehradun 248001, India; aswalsonali@gmail.com (S.A.); ankitkumarkoli88@gmail.com (A.K.); 2Department of Chemistry, Pt. Lalit Mohan Sharma Government Postgraduate College, Rishikesh 249201, India; ruchi_badoni26@yahoo.com; 3Department of Biotechnology, Faculty of Biomedical Sciences, Uttarakhand Ayurved University, Harrawala, Dehradun 248001, India; ashutosh_biotech2006@yahoo.com; 4Vice Chancellor, Uttarakhand Ayurved University, Harrawala, Dehradun 248001, India; akayu01@gmail.com; 5Department of Therapy Validation, Fraunhofer Institute for Cell Therapy and Immunology IZI, Perlickstr. 1, 04103 Leipzig, Germany; joerg.lehmann@izi.fraunhofer.de; 6Department of Phytochemistry, Faculty of Biomedical Sciences, Uttarakhand Ayurved University, Harrawala, Dehradun 248001, India

**Keywords:** Bufadienolides, cardiotonic, Indian squill, Kolkanda, traditional medicine

## Abstract

*Drimia indica* (Roxb.) Jessop (Asparagaceae) is a reputed Ayurvedic medicine for a number of therapeutic benefits, including for cardiac diseases, indigestion, asthma, dropsy, rheumatism, leprosy, and skin ailments. The present work aimed to critically and extensively review its traditional uses, phytochemistry, pharmacology, toxicology, and taxonomy together with the mechanisms of action of selected extracts of *D. indica*. A systematic literature survey from scientific databases such as PubMed, Scopus, and Web of Science as well as from some textbooks and classical texts was conducted. The plant, mainly its bulb, contains various bioactive constituents, such as alkylresorcinols, bufadienolides, phytosterols, and flavonoids. Various scientific studies have proven that the plant has anthelmintic, anticancer, antidiabetic, antimicrobial, antioxidant, and wound healing activities. The present work concludes that *D. indica* has the potential to treat various diseases, mainly microbial infections. This review also suggests that bufadienolides, flavonoids, and steroids might be responsible for its bioactive potential.

## 1. Introduction

The genus *Drimia,* belonging to the family *Asparagaceae*, is comprised of 99 accepted species [1], of which 9 species occur in India [2]. The members of this genus are mostly deciduous and rarely evergreen, with an underground bulb. These plants are distributed throughout Africa and Asia. Most of the species of this genus are found in southern Africa, mainly in semiarid regions with winter rainfall [3]. Various species of this genus, such as *D. maritima* (L.) Stearn, *D. elata* Jacq., *D. secunda* (B. Nord.) J.C. Manning & Goldblatt, and *D. indica* (Roxb.) Jessop are very popular in traditional medicine in different parts of the world [3,4].

*Drimia indica* (Roxb.) Jessop (syn. *Urginea indica* L.; *Scilla indica* Roxb.) is commonly known as Indian squill, true squill, or sea onion. In Asian countries, the plant is also called ban piaz or jungli piyaz [5]. *D. indica*, mainly its bulb, is an Ayurvedic medicine popularly known as Kolkanda or Ban Palandu that is used for the preparation of various medicinal products that have applications in healthcare, and it is also used as a biocide [6]. Phytochemical studies on the plant have revealed that alkaloids, flavonoids, phenols, and tannins are common in all parts, while steroids are only present in bulbs [7]. The bulbs also contain glycosides, quinones, resins, and saponins [8]. The plant is used in Ayurveda for respiratory disorders, skin diseases, dysmenorrhea, and intestinal worms. The plant has been studied in various biological activities and has exhibited potent antidiabetic, anticancer, antimicrobial, and cardiac effects. A systematic literature survey from scientific databases such as PubMed, Scopus, and Web of Science revealed that this is the first comprehensive and critical review comprised of the ethnobotany, phytochemistry, pharmacology, toxicology, and taxonomy of *D. indica*.

## 2. Materials and Methods

Literature based on the plant *D. indica* was thoroughly surveyed through online (scientific databases such as PubMed, Scopus, Web of Science, and Google Scholar) as well as offline (textbooks, magazines, and ancient classical texts) sources. For the online literature survey, the terms “*Drimia indica*”, “*Urginea indica*”, “*Scilla indica*”, “Indian squill”, “Kolkanda”, and “Ban Palandu” were searched. From online sources, about 175 articles/book chapters/webpages were obtained, and only relevant articles/papers were included in the present study. In this review, a critical compilation of the complete literature until the year 2018 has been included. The ethnobotany of the plant was mainly compiled from ancient Ayurvedic classics, including “Ayurvedic Materia Medica”.

## 3. Geographical Distribution

*D. indica* is endemic to India, Africa, and the Mediterranean region [6]. According to the “Flora of Zimbabwe” [9], it is found in open woodland, grassland, and rocky places and in margins of termite mounds in *zambesiacum* species in Botswana, Zambia, Zimbabwe, Mozambique, and South Africa. It is found from eastern tropical Africa west to Mauritania, from Senegal to Somalia and south to South Africa, and in northern Nigeria [10], India, Pakistan, Myanmar, Vietnam, and various other parts of the Indian subcontinent [11]. This plant is considered to be of “least concern” in the “Red List of South African Plants”, where it is popularly known as “secret lily” [12]. However, a recent report by Crouch et al. [13] revealed that plants corresponding to the type of *D. indica* are not with certainty known from the African continent. According to their preliminary molecular findings, the southern African material attributed to *D. indica* must be placed in *Vera duthiea*, and hence they excluded this plant from the South African plant species catalogue. Now, if this report is considered to be authentic, it can be said that the only occurrence of *D. indica* is in Asia, mainly in the Indian subcontinent.

## 4. Botanical Description

*Drimia indica* is a perennial herb having a bulbous geophyte. Earlier, this plant was known as *Scilla indica* or *Urginea indica*: However, at a later stage, Ansari and Raghavan [14] changed the name of Indian *Urginea* to *Drimia*. A similar species, *Drimia nagarjunae* (Hemadri & Swahari) Anand Kumar syn. *Urginea nagarjunae*, is considered to be a synonym of *D. indica* in the “Plant List” [1], which was first reported by Hemadri and Sasibhushan [15] from southern India. However, due to differences (e.g., having a thicker scape, flowers being closer together in the inflorescence and with tepals that are not reflexed), this species was considered accepted by the “World Checklist of Selected Plant Families” [16].

It is a small plant flourishing up to a height of 45 to 60 cm (Figure 1). The leaves are radical (appearing after the flower), linear, and acute. The bulbs are white with transparent outer scales and are 3–6 × 2–6 cm in size, consisting of fleshy coats that are thin and papery red or orange-brown in colour, enclosing each other completely. From the bulbs, smooth-edged leaves grow in the form of a rosette about 15–30 cm long and 1–2.5 cm wide and are lanceolate, linear, and acute. An erect flowering stem appears after the growth of the leaves. The flowers are small, pale green to white, and are very distant on slender, laxly flowered racemes. The flowers are arranged in dense clusters. The fruits are round, 1.5–2.0 cm in diameter with 6–8 flat seeds inside [8,17,18]. Cytogenetical studies have revealed the presence of 2*n* = 20 diploid chromosome numbers in *D. indica* [19].

## 5. Ethnobotany

In Ayurveda, the plant is popularly known as Kolkanda in the following Sanskrit (Roman) shloka (as per Raj Nighantu), which was originally written in Sanskrit by Pandit Narahari during the 17th century [20]:

Kolkandah katuschoshnah krimidoshvinashanah,Vantihrichchhardishmano vishdoshnivaranah.[Shloka No. 7/84]

The meaning of the above shloka is, “The Kolkanda is pungent and hot in nature. It suppresses the affection caused by germs/worms and also improves the conditions of nausea and vomiting. It antagonises the effect of poisons.”

On the other hand, according to the Bhavprakash Nighantu, which was originally written in Sanskrit by Acharya Bhav Mishra between the 15th and 16th centuries [21], the bulbs of the plant are expectorant, cardiostimulant, cardiotonic, diuretic, emetic, and nauseating. Its cardiovascular activity was found to be similar to that of foxglove (*Digitalis purpurea* L., family Plantaginaceae). The hot crushed mass of cocked bulbs is used on corns, whereas the powder is used for warts in Ayurveda. In addition, 10–15 drops of the bulb tincture (1:5), prepared with alcohol, is used for respiratory problems in children. According to Bhavprakash Nighantu, the suggested doses of its bulbs are 60–185 mg (dry powder), 30–60 drops (syrup/juice), and 5–30 drops (tincture).

According to Kirtikar [22], the bulbs of *D. indica* are stomachic, emmenagogue, anthelmintic, purgative, and alexiteric. They are useful for paralysis, bronchitis, asthma, dropsy, rheumatism, renal calculi, leprosy, skin diseases, headache, diseases of the nose, internal pains, and scabies in Indian traditional medicine. The bulbs are administered in the form of syrup (prepared from the bulb juice) as an expectorant in cases of bronchial catarrh and chronic bronchitis.

In Pakistan, aqueous extracts of the bulbs are used against inflammation due to injury, calculous affections, ulcers, sinusitis, and skin diseases such as leprosy [23]. In addition, they are used as an anthelminthic and a gastrointestinal stimulant to relieve constipation and indigestion [24]. Their use in treatments for coughing, bronchitis, asthma, paralytic affection, rheumatism, internal pain, and scabies is widely known in Asian countries, mainly in India and Pakistan. Crushed or sliced bulbs are applied under the soles of the feet to prevent burning sensations and are also used externally for removing corns and warts [25,26]. In the traditional medicinal system of Bangladesh, juice prepared from macerated leaves and fruits is taken on an empty stomach for acidity and dysentery [27].

In India, the plant is used as an alternative to *D. maritima* for bronchial and cardiac disorders. The crushed bulbs are topically used as a hair tonic for dandruff and seborrhea [28]. The bulbs are the source of a drug used as a cardiotonic, stimulant, expectorant, and diuretic. In the case of scarcity of bulbs, the leaves are also used as a medicine. The bulbs of *Dipcadi erythraeum* Webb & Berthel are sometimes used as a substitute and an adulterant in the formulations of *D. indica* [5]. A decoction of the fresh bulbs is used for the treatment of diabetes in various regions of the Indian Himalayas [29]. The dried powder of bulbs is used for rodent bites in India [2]. The juice obtained from the bulbs is given for kidney stones and to increase urine flow in some parts of central India [30].

A paste of bulbs with oil of *Madhuca longifolia* (Mahua) is applied to cure ulcers by the Gond tribe of Madhya Pradesh, India [31]. The bulb paste is also used against boils, tumours [32], joint pains [33], psoriasis, and many dermatological diseases [7]. The paste obtained from the leaf of *Achyranthes aspera* and the bulb of *D. indica* is used against scorpion stings in the Chattisgarh province of India [34].

## 6. Pharmacological Activities

*D. indica* has been studied for a variety of human ailments, mainly for use against microbial infections and cancers. In Ayurveda, it has long been used for a variety of disorders and is still prescribed by Ayurvedic practitioners. It is a well-known Ayurvedic medicine for skin diseases, intestinal worms, and respiratory disorders. Evidence-based research that has been carried out on this plant is given below, with research models, dose sizes, dose forms, possible bioactive constituents, and selected mechanisms.

### 6.1. Anthelmintic Activity

Helminthiasis, a macroparasitic disease in humans and animals, is caused by parasitic worms, mainly roundworms, hookworms, and whipworms [35]. The efficacy of various plants, including *D. indica*, in eliminating helminths has been reported. In the traditional system of medicine, bulbs of *D. indica* are used to treat helminthiasis. An aqueous extract obtained from the leaf, scape, and bulb was tested in vivo for anthelmintic potential against earthworm (*Pheretima posthuma*) due to its anatomical and physiological resemblance to human intestinal roundworm parasites. The extract, at 5 mg/mL, was found to be effective through the shown paralysis of earthworms at 41 min and death at 50 min. On the other hand, albendazole (5 mg/mL), a positive control in this study, showed paralysis at 92 min and the death of earthworms at 110 min [7]. The above study revealed that the crude extract of the bulbs was highly effective against earthworms and was equally potent to albendazole at a similar concentration. On the basis of this study, it can be suggested that active molecule/s of the plant certainly have a higher potential. However, this needs further advanced study to reach a final conclusion.

### 6.2. Antitumour Activity

An alcoholic extract of *D. indica* bulbs possessed activity against human epidermoid carcinoma of the nasopharynx in tissue cultures and against Friend virus leukaemia in mice at doses of 120–200 mg/kg [36,37]. A 29-kDa glycoprotein obtained from *D. indica* bulbs possessed potent anticarcinogenic activity by inhibiting angiogenesis through the inhibition of nuclear factor kappa B (NF-kB) nuclear translocation and activation and translocation of caspase activated DNase (CAD) into the nucleus, which is involved in apoptosis. The glycoprotein showed antitumour activity in terms of growth inhibition of mouse mammary carcinoma cells engrafted as an ascites tumour in a syngenic, immunocompetent mouse model at a dose of 75 µg, intraperitoneally per alternate day for six days. The protein exhibited antiangiogenic potential, including the inhibition of angiogenesis assays, decreased microvessel density count, and CD31 antigen staining in the peritoneum of treated mice. At a concentration of 10 µg, the glycoprotein also showed an inhibition of vascular endothelial growth factor (VEGF)-induced proliferation of human umbilical vein endothelial cells (HUVECs) in vitro. In comparison to the control, the glycoprotein showed 65% inhibition in the growth of HUVECs. The mechanism of angiogenesis inhibition involved reduced translocation of NF-kB into the nucleus, resulting in decreased expression of the *VEGFA* gene [38].

Likewise, a C-glycosyl flavone (5,7-dihydroxy-2-[4′-hydroxy-3′-(methoxymethyl) phenyl]-6-C-β-glucopyranosyl flavone) (Figure 2, structure no. **20**) isolated from *D. indica* bulbs was evaluated for its antitumour activity using the mouse model of Ehrlich ascites carcinoma (EAC). The compound, at a concentration of 25 µg/mL, was found to be active against EAC cells through a reduction in viability, restoration of cell cycle arrest, induction of apoptosis, inhibition of capillary formation, and reduced VEGF levels. These results were compared to 5-fluorouracil, a standard drug. The flavone was also found to reduce body weight, tumour volume, and the number of viable tumour cells and increase the survival of EAC-engrafted mice. It diminished pathological alterations induced by EAC cells and reduced the dissemination of EAC cells into the heart, kidneys, liver, and brain. The results also showed that it decreased serum thiobarbituric acid-reactive substance and lipid peroxidation and increased glutathione (GSH), dismutase (SOD), catalase (CAT), and glutathione peroxidase (GPX) [39]. So far, the anticancer activity of crude extracts and compound **20** (Table 1, Figure 1) has been proven against few cancer types. However, more cell lines and different cancer entities should be tested to evaluate activity against a broad variety of cancer types. In addition, normal cell lines should also be used to assess whether the cytotoxic effect is restricted to cancer cells or whether compounds from *D. indica* exhibit toxicity against normal cells, too. Those studies will be crucial in predicting the potential toxicity of *D. indica* in humans.

### 6.3. Antimicrobial Activity

Medicinal plants are known as an excellent source of antimicrobial drugs due to their potentially active phytochemicals. The bulb extracts of *D. indica* have been tested against a variety of microorganisms that are responsible for gonorrhoea, boils, joint pains, diarrhoea, gastrointestinal tract infections, pneumonia, skin infections, and urinary tract infections. Gonorrhoea, a sexually transmitted bacterial disease caused by *Neisseria gonorrhoeae*, is mainly found in vaginal fluid and discharge from the penis. There are several plants used in traditional medicine to treat sexually transmitted diseases that have also been validated scientifically [40]. The ethanol extract of bulbs was evaluated for antigonorrhoeal potential against 11 clinical isolates from patients with acute gonococcal urethritis and 7 WHO control strains of *N. gonorrhoeae*. The extract was found to be highly sensitive (51%–100% inhibition) against most of the isolates and strains. The results were compared to penicillin and ciprofloxacin [41]. Although the results were effective enough against *N. gonorrhoeae* and were comparable to standard antibiotics, e.g., penicillin and ciprofloxacin, studies on in vivo models are still missing. Hence, on the basis of in vitro studies, it is difficult to conclude whether a plant extract can be used as an antigonorrhoeal agent or not. An aqueous extract of bulbs showed potent inhibitory activity against gram-positive bacteria (*Bacillus brevis*, *B. licheniformis*, *B. subtilis*, *S. aureus*, and *Streptococcus aureus*), gram-negative bacteria (*Pseudomonas aeruginosa*, *Shigella flexneri*, and *Vibrio cholerae*), and fungus (*Candida kruse*), with an inhibition zone diameter (IZD) ranging between 19 and 28 mm at 200 µL [42].

A separate study by Panduranga et al. [43] revealed that the methanol extract of bulbs showed activity against *Escherichia coli, Staphylococcus aureus*, and *P. aeruginosa*, with an IZD range of 0.7–1.4 cm at 50, 100, and 150 mg/mL. The results were compared to levofloxacin at a concentration of 500 mg/disc, although this concentration looks to be too high for an in vitro study and may not be considered authentic, showing an IZD of 1.3 and 1.4 cm against *P. aeruginosa* and *E. coli*, respectively. On the other hand, pure methanol was used as a control in this study. The results were correlated with its traditional use in wound healing. Chittoor et al. [7] have also reported on the antimicrobial activity of an aqueous bulb extract against bacteria (*B. subtilis*, *S. aureus*, *E. coli*, and *P. aeruginosa*) and fungus (*Aspergillus niger* and *Candida albicans*). The extract was found to be effective against all tested bacteria (IZD of 12–13 mm) when compared to the reference drug ampicillin (IZD of 9–10 mm at 10 mg). The minimum inhibitory concentration (MIC) of the extract was found to be nearly 4–5 times lower than ampicillin. The MICs of the extract against *B. subtilis, S. aureus, P. aeruginosa*, and *E. coli* were recorded as 1.5, 3.15, 2.86, and 1.42 mg, respectively, whereas for ampicillin, the MICs were calculated as 8.2, 8.81, 10.86, and 8.4 mg, respectively. On the other hand, the activity of the extract against fungus pathogens was found to be comparatively lower (with an IZD of 21–22 mm) than with nystatin, which showed an IZD of 10–12 mm at 10 mg. The minimum inhibitory concentration (MIC) values of the extract were found to be 1.36 and 1.38 mg against *A. niger* and *C. albicans*, respectively, whereas the MICs of nystatin were recorded to be 0.25 mg against both fungi. It is noteworthy that that study did not unravel the bioactive constituents responsible for the antimicrobial activity. There is still a scope for future research to understand the exact mechanism(s) of action of antimicrobial activity with the help of in vivo and clinical studies.

Different extracts obtained from the root, stem, and leaves were found to be active against *Bacillus cereus*, *B. subtilis*, *S. aureus*, *Staphylococcus epidermidis*, *E. coli*, *Klebsiella pneumoniae*, *Proteus vulgaris*, *P. aeruginosa*, *A. niger*, and *C. albicans* at 2 mg/20 µL. Penicillin and streptomycin against bacteria and clotrimazole against fungi were used as standards at a dose of 10 µg/20µL. The methanol extract obtained from the roots showed the highest activity among all of the tested extracts, showing IZD values of 15.06, 14.33, and 12.33 mm against *B. cereus*, *S. epidermidis*, and *S. aureus*, respectively [8]. In the case of antifungal activity, an acetone extract of the root was found to be most effective, with IZDs of 14.06 and 13.26 mm against *C. albicans* and *A. niger*, respectively. The study was too preliminary, as the only zone of inhibition was measured against all of the pathogens. Minimum inhibitory concentrations should be recorded before using in vivo studies.

Apart from the crude extracts, three flavonoids—O-glycosyl flavanone (Table 1, Figure 2 (**23**)), O-glycosyl flavone (Table 1, Figure 2 (**24**)), and C-glycosyl flavone (Table 1, Figure 2 (**20**))—isolated from the methanol extract of *D. indica* bulbs showed antimicrobial activity against *S. aureus*, *B. subtilis*, *Rhizopus oryzae*, and *A. niger* at different concentrations ranging between 5 and 25 μg/mL [44]. The study further revealed that compound **20** was the most active and exhibited synergistic antibacterial activity with ciprofloxacin (fractional inhibitory concentration (FIC) index of 0.3 and 0.5 against *S. aureus* and *B. subtilis,* respectively) and synergistic antifungal activity with clotrimazole (FIC index of 0.3 and 0.48 against *R. oryzae* and *A. niger*, respectively). The compound (**20**) also increased ciprofloxacin-induced cytotoxicity against *S. aureus* (from 63% to 91%) and *B. subtilis* (from 56% to 89%) and clotrimazole-induced cytotoxicity against *R. oryzae* (from 36% to 49%) and *A. niger* (from 23% to 41%).

Besides, a 29-kDa glycoprotein of *D. indica* bulbs showed antifungal activity against various plant pathogens, *Fusarium oxysporum, Rhizoctonia solani, Sclerotium rolfsii*, and *Alternaria tenuissima*, with the maximum inhibition against *F. oxysporum* at 10 µg/well [2]. Another protein chitinase from the bulbs was also found to be active against *F. oxysporum* and *R. solani* [6].

### 6.4. Antioxidant Activity

Oxygen is utilised in the body to metabolise carbohydrate, fat, and protein, which results in producing energy. Due to the reactive nature of the oxygen atom, it is capable of becoming a part of potentially damaging molecules known as reactive oxygen species (ROS) including peroxides, superoxides, hydroxyl radicals, and singlet oxygen, which are the main cause of oxidative stress [45]. An antioxidant stabilises and deactivates ROS and other free radicals before they attack healthy cells. Carotenoids and flavonoids, together with other polyphenolics, are well-known naturally occurring antioxidant agents [46].

A methanol extract of *D. indica* bulbs exhibited in vitro 2,2-diphenyl-1-picryl-hydrazyl-hydrate (DPPH) radical scavenging activity (99.14%) at a concentration of 150 μg/mL [43]. Different concentrations ranging from 30 to 150 μg/mL of methanol extract were used in this in vitro study, which found that the extract showed activity varying from 98.10% to 99.14%. The methodology provided in this preliminary report, mainly the preparation of doses, was not clear. Moreover, there was no positive control used to compare the DPPH radical scavenging activity. On the other hand, the doses showing antioxidant effects were too high and could not be considered to be effective enough. Hence, this activity needs to be carried out further using standard protocols for its authenticity.

In a separate in vitro study, methanol, chloroform, and ethyl acetate extracts obtained from the bulbs showed in vitro antioxidant activity at IC_50_ values of 22.61, 23.00, and 24.98 µg/mL, respectively. The IC_50_ of the methanol extract was found to be equivalent to ascorbic acid (22.33 μg/mL) [47]. Concentrations of 10, 20, 40, and 60 μg/mL of different extracts were used to evaluate the DPPH scavenging activity. The results showed that the activity of the methanol extract was the highest, at 95.50%–97.57%, followed by chloroform (93.38%–95.91%) and ethyl acetate (86.40%–88.24%). This study suggested the presence of flavonoids in the plant extract but failed to correlate the activity with the responsible constituents.

Recently, Rajput et al. [48] reported that the bulbs of *D. indica* have phenolic and proanthocyanidin contents that are responsible for its antioxidant and free radical scavenging activities. This study used in vitro ferric reducing antioxidant power (FRAP) and DPPH assays and found a direct correlation between the concentration of total phenolics and antioxidant activity. This report mentioned *D. indica* and *D. nagarjunae* as different species, whereas according to the “Plant List” [1], these species are the same.

As the above reports were based on in vitro studies and were very preliminary in nature, the data obtained from such studies cannot be considered to be pharmacologically relevant, and the use of *D. indica* as an antioxidant is perhaps not safe until advanced in vivo and clinical studies are conducted. Moreover, this plant is neither used as a food nor as an antioxidant in traditional medicine: Hence, further studies are warranted to prove its use as a medicine against oxidative stress and other aging-related problems.

### 6.5. Antidiabetic Activity

In view of the traditional use of *D. indica* in diabetes, Gupta et al. [49] conducted an antidiabetic study of the ethanol extract of bulbs against streptozotocin-induced diabetes in rats. The extract (750 mg/kg and 1.5 g/kg of body weight) and the standard drug glibenclamide (10 mg/kg) were used orally for 14 days. The extract (at 1.5 g/kg) was found to contribute to considerable decreases in the blood glucose levels of diabetic rats within 120 min. In addition to a reduction in the blood sugar level, the extract was also found to reduce total cholesterol and triglyceride levels. Besides, the levels of high-density lipoproteins were found to improve compared to the group of untreated rats. The histopathological study revealed that the extract partially repaired the damaged cellular population of pancreatic islets in rats.

The effective dose, i.e., 1.5 g/kg, used in this study was too high to apply clinically: Bioassay-guided fractions or purified bioactives of the plant should be studied against diabetes. Moreover, the activity should not be considered to be good compared to glibenclamide, as its 10 mg/kg dose was used against 1.5 g/kg of the crude extract. This is the only scientific report available on the antidiabetic activity of this plant, and further studies using different approaches should also be carried out. The bioassay-guided fractions of the extract or isolated compounds of *D. indica* should be used for the activity.

### 6.6. Bronchodilator Activity

A bronchodilator is helpful for breathing trouble caused by narrowed or inflamed airways because it relaxes the lung muscles and dilates the airways. This medication is commonly used for the treatment of asthma and chronic obstructive pulmonary disease [50]. The *D. indica* bulb is used to treat chronic bronchitis and asthma in traditional medicine. To validate this claim scientifically, Bashir et al. [24] studied an aqueous ethanol extract of the bulb in rabbit tracheal and guinea pig atrial preparations. The extract inhibited contractions induced by carbachol (1 μM) and K^+^ (80 mM) in rabbit tracheae, similarly to dicyclomine. The results suggested the presence of Ca^2+^-channel blocking and anticholinergic mechanisms of the extract. The extract (0.01–1 mg/mL) increased the force of guinea pig atrial contractions without affecting their rate. This effect was perhaps mediated through the combined mechanism of an anticholinergic and Ca^2+^ antagonist accompanied by an inotropic effect.

This in vitro report gave an idea about the possible role of *D. indica* bulbs in bronchial diseases such as asthma and bronchitis. However, further studies with in vivo/clinical models are warranted before its use as a bronchodilator. Its purified fraction(s) or constituent(s) should be used to improve its efficacy and also to reduce dose sizes.

### 6.7. Gastrointestinal Stimulatory Activity

*D. indica* is traditionally used as a gastrointestinal stimulant to relieve constipation and indigestion. Therefore, the gastrointestinal stimulatory activity of an aqueous methanol extract of *D. indica* bulbs was studied using both in vitro and in vivo models [11]. The results showed that the extract (6 and 12 mg/kg) accelerated charcoal meal travel through the small intestine, similarly to a reference cholinergic drug, carbachol (10 mg/kg). The distance travelled by the charcoal meal in the extract-treated mice was 80.2% and 87.9% with 6 and 12 mg/kg, respectively, while with carbachol, the distance was 74.9%. Additionally, the extract showed a laxative effect at 10 mg/kg in atropine-induced mice, as reflected by an increase in the number of faeces, i.e., 4.3 and 9.2 at 6 and 12 mg/kg, respectively, over 6 h. This effect was found to be comparable to carbachol (10 mg/kg), which showed 11 faeces. In this study, we can see that the activity of the crude extract was almost similar to that of the standard at a similar or even a lesser dose. It is better to investigate the responsible chemical constituent(s) of the bulbs of *D. indica*. This could be great research in the field of novel drug discovery, particularly in treating gastrointestinal disorders.

The extract also showed an in vitro contractile effect in guinea pig ilea at 0.01–1.0 mg/mL as well as in rabbit jejuna at 0.01–0.3 mg/mL. At a concentration range of 0.01–5.0 mg/mL, the extract inhibited K^+^-induced contractions in rabbit jejuna and shifted the Ca^2+^ concentration–response curves to the right, which was comparable to the standard drug verapamil. The study suggested that the activity shown by *D. indica* was perhaps mediated through a cholinergic mechanism, which provides a rationale for its use in indigestion and constipation [11].

### 6.8. Anti-Inflammatory and Analgesic Activities

Nowadays, research on the discovery of natural drugs for pain and inflammation is of high interest due to the serious side effects, including gastrointestinal, renal, and respiratory problems, of currently available nonsteroidal anti-inflammatory drugs and opiates [51]. Although the plant *D. indica* is used for inflammation and internal pain in traditional medicine in Asian countries, scientific validation of the same has not been available. Thus, Rahman et al. [52] used an ethanol extract from the bulbs of *D. indica* to assess its anti-inflammatory and analgesic activities in Swiss albino rats. The study revealed that the ethanol extract, which was originally fractionated from the methanol extract of oven-dried material, at an oral dose of 1.5 g/kg exhibited significant anti-inflammatory activity against carrageenan-induced oedema in rats, with an inhibition range between 18.68% and 29.78% at 1–4 h when compared to the untreated control. On the other hand, the standard drug (ibuprofen) at an oral dose of 6 mg/kg inhibited oedema by a range between 23.07% and 41.84% 1–4 h post-treatment.

With a similar oral dose, i.e., 1.5 g/kg, the extract also exhibited analgesic activity in rats using a hot plate assay [52]. The extract increased hot plate pain perception in rats for up to 3 s in comparison to untreated rats. The activity could be considered to be poor in comparison to ibuprofen, which showed pain perception for up to 11 s for 4 h with an oral dose of 6 mg/kg. It is noteworthy to add here that the methodology and results reported by Rahman et al. [52] were poorly described, with many doubts. Hence, this report may not be considered to be authentic until supported by some further investigations.

The anti-inflammatory and analgesic activities shown by *D. indica* can be correlated with its traditional use against inflammation and internal pain. However, in this study, a crude ethanol extract was used: Studies on purified fractions/bioassay-guided fractions or isolated components are totally missing. Moreover, the oral dose, i.e., 1.5 g, used in the study was too high compared to the positive control, ibuprofen (6 mg/kg). Hence, as a crude drug, the extract cannot be considered to be potentially effective. Hence, further study is needed with purified fractions or constituents to evaluate its effectiveness against pain and inflammation.

### 6.9. Wound Healing Activity

A dichloromethane extract obtained from the bulbs of *D. indica* was evaluated for trauma healing activity in rats with skin trauma that was a 2-cm-long and 2-mm-deep cut. Local application of the extract at 100 mg/mL increased remodelling of the trauma area in comparison to the control group [10]. The study, however, failed to explain the bioactive compound responsible for the activity. Additionally, the mechanism of action was also poorly understood. In this regard, further research is needed to unravel the precise mechanism(s) of action and the bioactive constituent(s) present in the active extract/fraction.

## 7. Toxicity Studies

According to Bhavprakash Nighantu [21], the suggested doses of bulb for clinical use are 60–185 mg (dry powder), 30–60 drops (syrup/juice), and 5–30 drops (tincture). An acute toxicity study of ethanol extract of *D. indica* bulbs was tested on healthy rats at doses of 750 mg/kg and 1.5 g/kg of body weight. The study revealed that the extract did not cause lethality, toxic reaction, or behavioural, neurological, or autonomic changes in rats for up to seven days [49]. In addition, the acute and chronic toxicity of a C-glycosyl flavone (Table 1, Figure 1 (**20**)) isolated from the bulbs of *D. indica* was also evaluated in mice. The study revealed that the flavone exhibited a safety profile in toxicity experiments [39]. In large doses, however, the plants, mainly their bulbs, were emetic and cathartic and could cause cardiac depression [5]. A classical Ayurvedic text has also revealed that although the bulbs of the plant control nausea and vomiting, large doses can produce vomiting and should not be used as such without processing [20]. Deepak et al. [2] have also mentioned that the plant is not consumed by either humans or animals due to its toxicity. Hence, the oral administration of *D. indica* in any form may be a matter of serious concern, and further toxicity studies are warranted before its use as an oral medicine.

## 8. Secondary Metabolites

The plant *D. indica* contains a variety of bioactive constituents, including alkaloids, sterols, flavonoids, anthraquinones, and coumarins [53,54,55]. Many other constituents, such as tannins, quinones, resins, saponins, phenolics, protein, and carbohydrates such as sucrose, maltose, fructose, and galactose, have also been reported in different parts of the plant [11,24,43]. Chromatographic techniques such as column chromatography and high performance liquid chromatography have been used to separate the compounds and characterise them with the help of chemical and advanced spectroscopic techniques, including 2D nuclear magnetic resonance and high-resolution mass spectrometry studies. 

Mikail et al. [10] found that a total yield of cichoriol A and urgineol A–C in a dichloromethane extract of the bulbs was 0.11%, whereas the yield of urgineol D–F was 0.14%. They also found that oleic acid, linoleic acid, and squalene were the major lipid constituents of the dichloromethane extract. The chemical constituents isolated from *D. indica* are depicted in Table 1, and their chemical structures are drawn in Figure 2.

## 9. Future Perspectives and Recommendations

The availability of *D. indica* is seasonal and comparatively less, as its major availability is on the Indian subcontinent, mainly at mid-high altitudes (around 1500 m to 3000 m) in the Himalayan region. Hence, its demand as a medicine is rather difficult to fulfil. Thus, on an industrial scale, the cultivation of this wild plant is highly essential. Nowadays, there is an increasing interest of scientists in the in vitro cultures of many valuable medicinal plants to fulfil their market demand: This includes Agastya (*Sesbania grandiflora* (L.) Pers.), a well-known remedy for bronchitis, headaches, fevers, and anaemia [58]. In addition, the genetic material of such plants can also be used in culture techniques for their growth and multiplication, as well as for the improvement of their secondary metabolites. Using these biotechnological procedures, several important medicinal herbs, such as Hastidanti (*Baliospermum montanum* (Willd.) Müll.Arg.), Dhataki (*Woodfordia fruticosa* (L.) Kurz), and Ashwagandha (*Withania somnifera* (L.) Dunal), have been productively grown in past years [59].

Interestingly, *D. indica* is a source of proscillaridin A (Table 1, Figure 2 (**9**)), which has been clinically proven as a drug for congestive heart failure and cardiac arrhythmia [60]. Moreover, due to the similarity of constituents such as the cardiac glycosides in *D. indica* to those of scillaren A and B of the Mediterranean squill (*D. maritima*) (very little difference), the plant is considered to be a good substitute for *D. maritima*, which is used worldwide as a cardiac drug [61]. 

Various other constituents, such as bufadienolides, flavonoids, and steroids, are also present in this plant, which might be responsible for its bioactive potential. Hence, *D. indica* can be used as the source of a new drug either in the form of a crude extract or as individual compounds isolated from the plant. In Ayurveda, this plant has also been suggested to be as effective as *Digitalis purpurea* L. (Plantaginaceae), particularly against heart-related problems, and can be used as a substitute. Since *D. indica* acts more rapidly and less effectively than digitalis, its use as a substitute cannot be encouraged due to its irritating effect, poor absorption, and requirement of higher doses [37]. However, further research based on drug delivery and development can improve its absorption and reduce the dose size, which can help in its acceptance as a drug. There are several methods to improve the absorption of drugs with low permeability into gastrointestinal tract (GIT), including many drug delivery devices and systems using different formulations and excipients [62]. Particularly in the case of herbal remedies, the development of polyherbal formulations [63] or the use of nanoparticles [64] can be considered to be the best ways to improve efficacy and bioavailability.

Although scientifically not proven, a few reports have stated that the plant is toxic in nature [5,20]. The toxicity of this plant can be observed through in vitro studies on different cell lines or in vivo exposure in different animal models [65]. Since preclinical toxicity studies help in initiating the clinical evaluation of a product, further studies must be carried out with the plant to evaluate its toxicity in relation to its efficacy in a broad variety of disease models in vivo.

Most of the available literature has been based on the biological activity of crude extracts of the plant, and studies with purified compounds have been very few. On the other hand, responsible chemical constituents showing activity have still not been unravelled. Moreover, clear mechanisms of action, as well as structure–activity relationships, are totally missing in the available literature. Hence, further research is needed to find out the possible mechanisms of action and structure–activity relationships in purified bioactive compounds of *D. indica*. The structure of several compounds implies that the aryl hydrocarbon receptor might be involved in some of the observed pharmacological effects of *D. indica* extracts, which is a matter for further studies.

## 10. Conclusions

This study concludes that the bulbs of *D. indica* can be considered to be safe within a range of accepted doses, i.e., 60–185 mg (dry powder), 30–60 drops (syrup/juice), and 5–30 drops (tincture), because the plant has been used in Ayurveda for many years. Traditionally, the plant has also been used topically for wound healing, fungal infections, and removing corns and warts. Hence, topical use of the plant can be easily accepted, as there has been no toxicity report on it when it has been applied externally.

Besides, various extracts of *D. indica* have shown anthelmintic, anticancer, antimicrobial, antioxidant, antidiabetic, bronchodilator, gastrointestinal stimulatory, anti-inflammatory, analgesic, and wound healing activities in different in vitro and in vivo models. However, clinical studies are totally missing on this important medicinal plant, although it has been used in Ayurveda for the past many centuries. The reasons might be its poor and seasonal availability, poor toxicity information, and also the availability of substitute plants, e.g., *Drimia maritima* (L.) Stearn and *Digitalis purpurea* L. Hence, there is still a good scope for future research based on clinical trials of this plant.

## Figures and Tables

**Figure 1 medicina-55-00255-f001:**
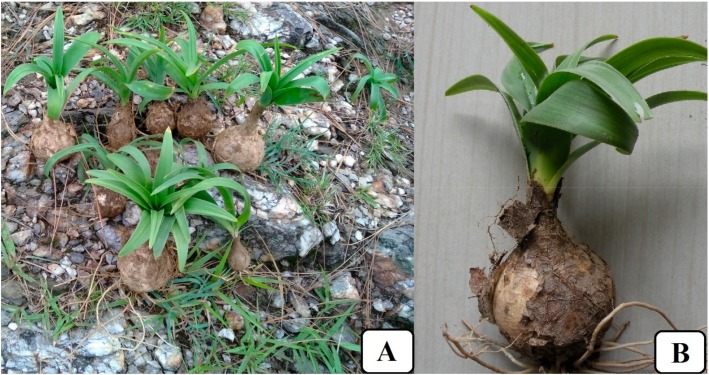
*Drimia indica*: (**A**) freshly harvested plants from their natural habitat; (**B**) a single whole plant.

**Figure 2 medicina-55-00255-f002:**
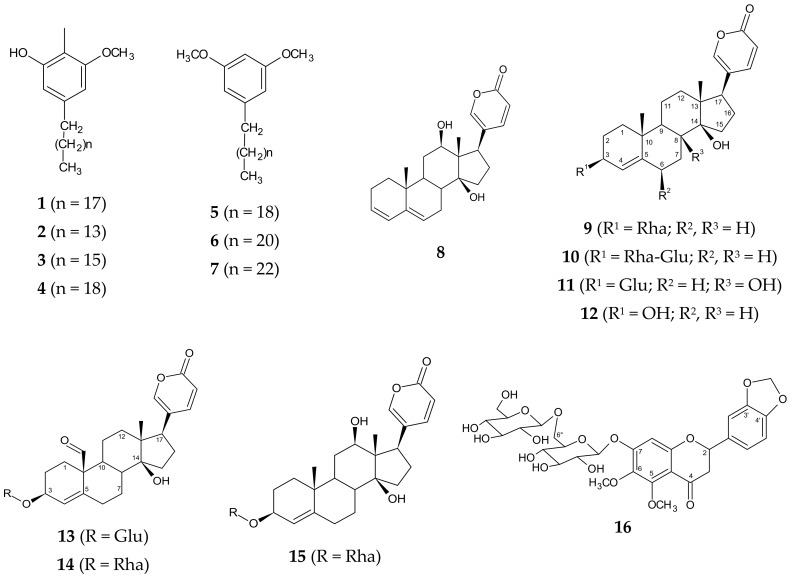
Chemical structures of compounds reported from *Drimia indica*.

**Table 1 medicina-55-00255-t001:** Chemical constituents reported from various parts of *Drimia indica*.

Chemical Constituents	Plant Part	Structure No.	Reference
**Alkylresorcinols**			
Cichoriol A	Bulb	**1**	[10]
Urgineol A	Bulb	**2**	[10]
Urgineol B	Bulb	**3**	[10]
Urgineol C	Bulb	**4**	[10]
Urgineol D	Bulb	**5**	[10]
Urgineol E	Bulb	**6**	[10]
Urgineol F	Bulb	**7**	[10]
**Bufadienolides**			
Anhydroscilliphaeosidin	Bulb	**8**	[53]
Proscillaridin A/Scillarenin-3-O-α-L-rhamnoside	Bulb and root	**9**	[53,56]
Scillaren A/Scillarenin-3-O-α-L-rhamnosido-β-d-glucoside	Bulb and root	**10**	[53,56]
Scilliphaeoside	Bulb and root	**11**	[53]
Scillarenin	Bulb and root	**12**	[53]
Scilliglaucosidin-3-O-β-d-glucoside	Bulb	**13**	[56]
Scilliglaucosidin-3-O-α-L-rhamnoside	Bulb	**14**	[56]
Scilliphaeosidin-3-O-α-L-rhamnoside	Bulb	**15**	[56]
**Flavonoids**			
5,6-Dimethoxy-3′,4′-dioxymethylene-7-O-(6″-β-d-glucopyranosyl-β-d-glucopyranosyl) flavanone	Bulb	**16**	[54]
5,4′-Dihydroxy-3-O-α-L- rhamnopyranosyl-6-C- glucopyranosyl-7-O-(6″-cis-*p*-coumaroyl-β-d- glucopyranosyl) flavone	Bulb	**17**	[54]
5,4′-Dihydroxy-3-O-(2′′′″-β- glucopyranosyl-α-L-rhamnopyranosyl)-6-C-glucopyranosyl-7-O-(6″-cis-*p*-coumaroyl-β-d- glucopyranosyl) flavone	Bulb	**18**	[54]
Apigenin	Leaf	**19**	[7]
5,7-Dihydroxy-2-[4′-hydroxy-3′-(methoxymethyl) phenyl]-6-C-β-glucopyranosyl flavone	Bulb	**20**	[44]
Kaempferol	Scape	**21**	[7]
Luteolin	Bulb	**22**	[7]
5,7,4′-Trihydroxy-3′-methoxy-7-O-(6″-α- L-rhamnopyranosyl-β-d-glucopyranosyl flavone	Bulb	**23**	[44]
5,4′-Dihydroxy-3′-methoxy-7-O-β-d- glucopyranosyl flavone	Bulb	**24**	[44]
Quercetin	Bulb, leaf, and scape	**25**	[7]
**Phenolic compounds**			
Coumarin	Bulb	**26**	[7]
Salicylic acid	Bulb	**27**	[7]
**Phytosterols**			
Campesterol	Bulb, leaf, and root	**28**	[57]
β-Sitosterol	Bulb, leaf, and root	**29**	[57]
Stigmasterol	Bulb, leaf, and root	**30**	[57]

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
