# Peer review of "Drimia indica: A Plant Used in Traditional Medicine and Its Potential for Clinical Uses"

_medicina, 2019, doi:10.3390/medicina55060255_

Round 1

Reviewer 1 Report

In the manuscript, authors present review comprised of ethnobotany, phytochemistry, pharmacology, toxicology, and taxonomy of D. indica.

The subject of the work is interesting. However, I have doubts whether the work could be published in this form.

The opinion of the authors is not critical enough. Most of the reports on antioxidant, anticancer, antidiabetic, antimicrobial, anthelmintic activity of the species base conclusions on research involving very high doses of raw material, which would most probably not be possible to achieve in the human body. In addition, they would not be safe for human. For example, only a dose of 1.5 g/kg exhibited significant anti-inflammatory activity or MICs of extract was 1.5 - 3 mg / ml, when antibiotics have MIC values expressed in µg micrograms.

In my opinion, a deeper analysis of the results obtained in the quoted publications is required. Particularly because the part concerns works that have been published in little-known magazines without a factor of influence.

Particularly questionable is the alleged high security of the received raw material in relation to the compounds contained in them. Cardiac glycosides, including presented in the species, bufadienolides, are very strong drugs, plants containing them have a narrow therapeutic index and could give a poisoning, even lethal effect. Meanwhile, the doses indicated for various applications are very high: for example, against leukemia at the dose of 120-200 mg / kg, i.e. a dose of about 10-15 g for humans or the effective antidiabetic dose 1.5 g/kg, i.e. about 110 g for a human.

In my opinion, most of the presented studies show that there is no confirmation for many uses of the raw material in the conducted research, and the authors only to a certain extent emphasize these doubts.

The authors have no influence on the results of the research, but their review should be more critical and  better discussing the over-interpretation of results the described research.

In addition, the work is missing a quantitative content of bioactive compounds in the plant. The fact that the compound has been identified in the raw material does not mean that it is present in essential quantity for activity. These data on particular groups of compounds or their dominant representatives are definitely missing at work.

Additionally, the Conclusion section is too long and there are no reflections on future prospects for the use of the species.

Author Response

Comment 1: In the manuscript, authors present review comprised of ethnobotany, phytochemistry, pharmacology, toxicology, and taxonomy of D. indica. The subject of the work is interesting. However, I have doubts whether the work could be published in this form.

Response: Thank you for your valuable comments. As per your suggestion, we have revised the manuscript and the changes made in the manuscript are highlighted with yellow font.

Comment 2: The opinion of the authors is not critical enough. Most of the reports on antioxidant, anticancer, antidiabetic, antimicrobial, anthelmintic activity of the species base conclusions on research involving very high doses of raw material, which would most probably not be possible to achieve in the human body. In addition, they would not be safe for human. For example, only a dose of 1.5 g/kg exhibited significant anti-inflammatory activity or MICs of extract was 1.5 - 3 mg / ml, when antibiotics have MIC values expressed in µg micrograms.

Response: Indeed, the doses used to evaluate certain activities are too high and cannot be considered as safe for in vivo and also for clinical studies. We have critically described the shortcomings of the studies and given our own opinion for further research. The critical discussion on the anticancer, antidiabetic, antimicrobial, antioxidant activities is given/ added and highlighted with yellow font.   

Comment 3: In my opinion, a deeper analysis of the results obtained in the quoted publications is required. Particularly because the part concerns works that have been published in little-known magazines without a factor of influence.

Response: As we mentioned in the abstract as well as in the introduction part that this is a comprehensive review on the D. indica. We have included articles from Scopus, PubMed, and Google scholar, as well as from some little-known magazines which are also available online. We understand that the work mentioned in these magazines may not authentic enough but without further research/ evidence, we cannot ignore their facts. Hence, we although included these reports but parallelly given a critical viewpoint on their shortcomings. I would like to add here that, sometimes a regional/ little-known research/ report can give worthy information or a lead to conduct an advance study. I hope you will consider my request to remain such reports in the paper. 

Comment 4: Particularly questionable is the alleged high security of the received raw material in relation to the compounds contained in them. Cardiac glycosides, including presented in the species, bufadienolides, are very strong drugs, plants containing them have a narrow therapeutic index and could give a poisoning, even lethal effect. Meanwhile, the doses indicated for various applications are very high: for example, against leukemia at the dose of 120-200 mg / kg, i.e. a dose of about 10-15 g for humans or the effective antidiabetic dose 1.5 g/kg, i.e. about 110 g for a human.

Response: Based on the available research/ reports on the plant, we have given a separate section for toxicity studies. This contains following text.

“According to Bhavprakash Nighantu, the suggested doses of its bulbs for clinical uses are 60-185 mg (dry powder), 30-60 drops (syrup/ juice) and 5-30 drops (tincture). The acute toxicity study of ethanol extract of D. indica bulbs was tested on healthy rats at the doses of 750 mg/kg and 1.5 g/kg body weight. The study revealed that the extract did not cause lethality, toxic reaction or behavioural, neurological, and autonomic changes in rats up to 7 days. In addition, the acute and chronic toxicity of a C-glycosyl flavone isolated from the bulbs of D. indica was also evaluated in mice. The study revealed that the flavone exhibited safety profile in toxicity experiments. In large doses, however, the plants, mainly their bulbs, are emetic and cathartic and may cause cardiac depression. The classical Ayurvedic text also revealed that although the bulbs of the plant control nausea and vomiting, its large doses can produce vomiting and should not be used as such without processing. The plant is not consumed either by human or by animals due to its toxicity. Hence, the oral administration of D. indica in any form may be the matter of a serious concern, and further toxicity studies are warranted before its use as an oral medicine.”

Comment 5: In my opinion, most of the presented studies show that there is no confirmation for many uses of the raw material in the conducted research, and the authors only to a certain extent emphasize these doubts. The authors have no influence on the results of the research, but their review should be more critical and better discussing the over-interpretation of results the described research.

Response: Whatever described in this review paper is evidence-based. We carefully emphasized all areas in which the literature was available with a critical analysis. In some cases, the possible mechanism of action is fully described, for example bronchodilator and Gastrointestinal stimulatory activities whereas if the mechanism of action for a particular activity is not clear, it is also mentioned in the text.  

Comment 6: In addition, the work is missing a quantitative content of bioactive compounds in the plant. The fact that the compound has been identified in the raw material does not mean that it is present in essential quantity for activity. These data on particular groups of compounds or their dominant representatives are definitely missing at work.

Response: Information related to the yield of the selected constituents has been included under secondary metabolites section.

Comment 7: Additionally, the Conclusion section is too long and there are no reflections on future prospects for the use of the species.

Response:  As per the suggestion, the section is splitted into two parts i.e. Future perspectives and recommendation, and conclusion.

Reviewer 2 Report

Dear authors

In the manuscript medicina-479692, the authors present the traditional uses and the scientific studies on extracts (in vitro studies) as well as on its phytochemical composition, toxicity and pharmacology that could support the traditional use of Drimia indica species.

Overall the manuscript has interest, the subject falls within the scope of the Medicina, has novelty and could attract the interest from a great number of readers.

I have only some suggestions to improve, even more, the quality of the manuscript and some correction to minor points.

-          I believe that the manuscript can be improved especially at the level of organization.

For example, considering the methodology used in the phytochemical and pharmacological characterization of plants, I believe it would be more logical the following organization after point 4:

5.       Ethnobotany

6.       Pharmacological activity

7.       Toxicity

8.       Phytochemical

9.       Conclusions and recommendations

-          I also consider that it is very relevant that the authors do not limit themselves to saying that an extract or compound data is active. They should indicate their IC50 (or concentration at which activity is achieved) as well as the activity of a positive control under the same experimental conditions. The authors, in some cases, present these data. However, they should be more consistent and present in all cases. Where such data do not exist, authors should give to the readers such indication or otherwise delete the reference from this manuscript. In fact, if there is a published paper on the biological activity of a compound or extract, and the activity of a positive control in that work is not shown, there is no way to gauge the accuracy of that study. Thus, it is an incomplete study that a critical review does not must give credit. Only by using a positive control for comparison will the reader be able to have an accurate picture of the level of activity displayed, contributing to increase the impact of the manuscript.

-          Please, check carefully the chemical name of all compounds.

In the chemical name of compound 16 there is an error that originates from the article Sultana et al., 2010. The correct name is 5,6-dimethoxy-3′,4′′-dioxymethylene-7-O-(6′′-β-D-glucopyranosyl-β-D-glucopyranosyl) flavanone, instead of 5,6-dimethyoxy-3′,4′′-dioxymethylene-7-O-(6′′-β-D-glucopyranosyl-β-D-glucopyranosyl) flavanone.

Compound 17: Has the 6’’-para-coumaroyl substituent a cis or trans double bond configuration? The structure shows a cis configuration, but the chemical name doesn’t have the cis or trans indication. Please clarify.

In the chemical name of the glycosides, the letter D or L must be written with the size 2 points below the rest of the text. For example, in the name of compounds 23 and 24. Please check all the chemical names.

-          In order to promote the readability of Figure 2, I suggest that the chemical names of the compounds be removed from this figure, leaving only the compound number and the identification of the substituents R.

-          Line 36: the “The plant List” database, must be correctly cited in the references list.

-          Line 43: Drimica indica have a lot of synonym. Are these two the only that have ethnopharmacological relevance? If yes, it is OK. If not, please add another significant botanical synonym.

-          Authors should carefully check the reference list and formatting it following the author instructions of Medicina journal. The references must be numbered in order of appearance in the text instead Author name and year.

And in the final reference list it should be:

1.        Author 1, A.B.; Author 2, C.D. Title of the article. Abbreviated Journal Name Year, Volume, page range.

2.        Author 1, A.; Author 2, B. Title of the chapter. In Book Title, 2nd ed.; Editor 1, A., Editor 2, B., Eds.; Publisher: Publisher Location, Country, 2007; Volume 3, pp. 154–196.

Author Response

Comment 1: In the manuscript medicina-479692, the authors present the traditional uses and the scientific studies on extracts (in vitro studies) as well as on its phytochemical composition, toxicity and pharmacology that could support the traditional use of Drimia indica species. Overall the manuscript has interest, the subject falls within the scope of the Medicina, has novelty and could attract the interest from a great number of readers. I have only some suggestions to improve, even more, the quality of the manuscript and some correction to minor points.

Response: Thank you for evaluating this manuscript in such a beautiful way and suggesting some valuable comments to improve the manuscript. As per your suggestion, we have revised the manuscript and the changes are highlighted with yellow font.  

Comment 2: I believe that the manuscript can be improved especially at the level of organization.

For example, considering the methodology used in the phytochemical and pharmacological characterization of plants, I believe it would be more logical the following organization after point 4:

5.       Ethnobotany

6.       Pharmacological activity

7.       Toxicity

8.       Phytochemical

9.       Conclusions and recommendations

Response: As per suggestion, the manuscript is reorganized and the phytochemical part has been moved after toxicity studies.

Comment 3: I also consider that it is very relevant that the authors do not limit themselves to saying that an extract or compound data is active. They should indicate their IC50 (or concentration at which activity is achieved) as well as the activity of a positive control under the same experimental conditions. The authors, in some cases, present these data. However, they should be more consistent and present in all cases. Where such data do not exist, authors should give to the readers such indication or otherwise delete the reference from this manuscript. In fact, if there is a published paper on the biological activity of a compound or extract, and the activity of a positive control in that work is not shown, there is no way to gauge the accuracy of that study. Thus, it is an incomplete study that a critical review does not must give credit. Only by using a positive control for comparison will the reader be able to have an accurate picture of the level of activity displayed, contributing to increase the impact of the manuscript.

Response: Biological activities of Drimia indica and its isolates based on the available literature have been described with critical analysis. We have included the doses used and the activities are also compared with the standards. In some studies, the standards have not used which is mentioned in the text. The doses/ concentrations are highlighted with yellow font.

Comment 4: Please, check carefully the chemical name of all compounds.

In the chemical name of compound 16 there is an error that originates from the article Sultana et al., 2010. The correct name is 5,6-dimethoxy-3′,4′′-dioxymethylene-7-O-(6′′-β-D-glucopyranosyl-β-D-glucopyranosyl) flavanone, instead of 5,6-dimethyoxy-3′,4′′-dioxymethylene-7-O-(6′′-β-D-glucopyranosyl-β-D-glucopyranosyl) flavanone.

Compound 17: Has the 6’’-para-coumaroyl substituent a cis or trans double bond configuration? The structure shows a cis configuration, but the chemical name doesn’t have the cis or trans indication. Please clarify.

In the chemical name of the glycosides, the letter D or L must be written with the size 2 points below the rest of the text. For example, in the name of compounds 23 and 24. Please check all the chemical names.

Response: Thank you for notifying this major correction. The name of compound 16 is corrected. In the structure 17 and 18, there is cis configuration and now given in the text. The letters D or L have been resized 2 points below the rest of the text.  

Comment 5: In order to promote the readability of Figure 2, I suggest that the chemical names of the compounds be removed from this figure, leaving only the compound number and the identification of the substituents R.

Response: The names of the compounds in the table have been removed.  

Comment 6: Line 36: the “The plant List” database, must be correctly cited in the references list.

Response: The reference is corrected.  

Comment 7: Line 43: Drimica indica have a lot of synonym. Are these two the only that have ethnopharmacological relevance? If yes, it is OK. If not, please add another significant botanical synonym.

Response: Only Drimia indica and Urginia indica have ethnopharmacological relevance.   

Comment 8: Authors should carefully check the reference list and formatting it following the author instructions of Medicina journal. The references must be numbered in order of appearance in the text instead Author name and year.

And in the final reference list it should be:

1. Author 1, A.B.; Author 2, C.D. Title of the article. Abbreviated Journal Name Year, Volume, page range.

 2. Author 1, A.; Author 2, B. Title of the chapter. In Book Title, 2nd ed.; Editor 1, A., Editor 2, B., Eds.; Publisher: Publisher Location, Country, 2007; Volume 3, pp. 154–196.

Response: The references are now given as per Medicina journal guidelines.

Round 2

Reviewer 1 Report

The authors' responses are satisfactory to me, and the manuscript has been improved. Now,

in my opinion the article can be accepted for publication.

Author Response

n/a
